# Electric Vehicle Charging Infrastructure Policy Analysis in China: A Framework of Policy Instrumentation and Industrial Chain

Xin Wang [1], Jinfeng Wang [2,*], Chunqiu Xu [3], Ke Zhang [3] and Guo Li [1]

1    School of Politics and Public Administration, Zhengzhou University, Zhengzhou 450001, China
2    China Institute of FTZ Supply Chain, Shanghai Maritime University, Shanghai 201306, China
3    School of Management, Zhengzhou University, Zhengzhou 450001, China
*    Correspondence: wangjinfeng@shmtu.edu.cn

**Abstract:** As a strategic guarantee for the rapid development of electric vehicles, the construction and development of electric vehicle charging infrastructure (EVCI) is closely related to the industrial policies formulated by the government. This paper takes policy texts relevant to EVCI in China since 2014 as the research materials, taking policy instruments and the industrial chain as analysis dimensions. Policy content analysis is conducted to explore the EVCI policy content, structure characteristics of policy instruments, and evolution characteristics of EVCI policy in China. Our research reveals that China's EVCI policy system is relatively perfect, but the use of policy instruments is not balanced and, in particular, is not coordinated with the EVCI industrial chain they supported. In this regard, the government should pay more attention to the use of demand-side policy instrument to enhance the driving force for the development of the EVCI industry. With more scientific and reasonable arrangement of the distribution and implementation of policy instruments in the EVCI industrial chain, the benign development of China's EVCI industry can be promoted. This research contributes to strengthening the management and policy instrumentation of the central Chinese government, in order to support the realization of good governance of EVCI and the new energy vehicle development.

**Keywords:** electric vehicle charging infrastructure; policy instrumentation; industrial chain; policy content analysis

## 1. Introduction

In the face of the severe resource and environmental crises, it has become the obvious choice for many countries to make greater efforts to encourage research and development into clean energy on the basis of the economical use of conventional energy [1]. Inevitably, vigorously and continuously promoting the development of electric vehicles is an effective path for optimization of China's energy structure.

The development of the electric vehicle charging infrastructure (EVCI) industry is crucial for the large-scale promotion of new energy vehicles [2]. Under the promotion of relevant national policies, China's EVCI industry has developed rapidly in recent years, with the scale of construction expanding and the gap between vehicle–pile ratios gradually narrowing. However, the current number of charging piles is far from both the actual demand and the targets set by the relevant authorities. According to the New Energy Vehicle Industry Development Plan (2021–2035), by 2025, the number of new energy vehicles sold in China will account for a quarter of the overall number of vehicles sold; furthermore, by 2030, pure electric vehicles will be the mainstay of vehicles sold [3]. It is easy to foresee that, as an important supporting industry for new energy vehicles, the demand for EVCI will surely grow geometrically and, thus, China's EVCI industry will enter the fast lane of rapid development.

Policy support is a strong guarantee to promote the development of an industry. Therefore, it is of great theoretical and practical significance to conduct an in-depth study of EVCI policies. For example, what policies have been put in place in China to encourage the development of EVCI industry? What policy instruments have been used? What are the characteristics and changing trends of these policy instruments and which industry chain parts are they applied to? To answer these questions, in this paper, we conduct policy content analysis and construct a "policy instruments–industrial chain" analysis framework, of in order to analyze the central EVCI policies in China released from 2014 to 2021. This study aims to provide scientific policy-making reference for the government to promote the rapid and stable development of the EVCI industry.

## 2. Literature Review

The rapid development of electric vehicles has caused the EVCI industry to gradually become a hotspot of theoretical research. The existing literature has mainly focused on the development of the EVCI industry and related policy research [4].

### 2.1. Development of EVCI Industry

Previous research on the EVCI industry has mainly focused on client demand forecasting, location selection and layout optimization, and operation and business models for EVCI. To address the problem of planning site selection in the pre-development phase of the EVCI industry, Cao et al. established a user travel cost location model by predicting the charging load at the planned site, while using a genetic algorithm to determine an optimal solution for the location of charging stations [5]. Liu proposed a strategy for planning electric vehicle charging stations which takes into account construction costs and driver satisfaction by examining the planning of electric vehicle charging stations along German motorways [6]. As such, these authors have studied the impacts of different factors on the planning and layout of charging stations from different perspectives.

With the unbalanced utilization problem caused by the rapid development of the EVCI industry, some scholars have studied how to predict user demand to achieve a good match between charging infrastructure and user demand, in order to improve the idle problem of charging infrastructure [7]. Chen used the fuzzy demand–profit model to optimize the utilization problem of EV charging infrastructure to effectively improve the efficiency of new energy supply [8]. Unlike Chen's study using the demand–profit model, Ye studied the demand of users from a multi-objective selection perspective, in order to enable the user electricity needs to be met with minimal charging and generation costs [9]. It may seem that these authors conducted research from different perspectives but, in fact, both of them studied the relationship between resources and the needs of the users. With the development of charging technologies and products, fast charging infrastructure can significantly reduce charging times and improve the charging efficiency. A comprehensive framework for urban fast charging infrastructure is being constructed, in order to address the issue of mileage anxiety [10].

In addition, many scholars have conducted studies on the operation and business models related to the EVCI. For example, Mercan et al. constructed a hybrid operating model of charging stations incorporating solar and energy storage to optimize the dispatch of EV charging units and predicted the benefits provided to potential operators by assessing the environmental and economic benefits [11]. Zhang et al. proposed a mobile charging solution to the charging problem in urban areas, and investigated the economic competitiveness of mobile charging based on a comparison of the convenience and cost of using traditional fixed charging posts and mobile charging posts [12]. Sarker et al. proposed a framework for optimizing the offer/bidding strategy for a combination of integrated charging stations and energy storage systems, and the results showed that the framework can provide cost savings for integrated charging stations [13].

In summary, although the existing studies have covered a wide range of topics, including all aspects of the EVCI industrial chain, they are still inadequate, when compared to

studies related to the new energy vehicle industry. Furthermore, these studies explored the development of the EVCI industry at a micro level, neglecting to construct and analyze the links between the development of the industry and related policies. Such links deserve to be explored especially in China, the country with the largest charging network in the world. In order to understand the relationship between government policy and industrial development deeply and comprehensively, in this paper, we match government measures with the industry chain activities of EVCI. On this basis, specific support measures that can be taken by the government for promoting the development of EVCI industry are derived.

### 2.2. EVCI Policy Research

The related literature has mainly focused on the relationship between EVCI policies, industrial development, and the impact of industrial policies on related industries. Helmus et al. evaluated both strategic deployment and demand-driven rollout strategies commonly used for EVCI globally, and they found that the perfect implementation strategy would be a policy mix of demand-driven and strategic deployment [14]. Cansinao et al. analyzed the policy instruments used to promote electric vehicles in the EU28 and concluded that, in addition to fiscal incentives for government procurement and support for research and development (R&D) projects, the most common measure used to promote electric vehicles was support for EVCI installation, followed by free parking for electric vehicles [15]. In contrast, Baumgarte et al. argued that currently available policy measures, such as investment subsidies or exemptions from electricity taxes, do not contribute significantly to the widespread expansion of fast charging infrastructure, while charging demand has a greater potential to contribute [16]. If there are few charging facilities, policy subsidies are almost ineffective and charging companies are reluctant to build charging infrastructure without incentives; in this case, the government must take the initiative to promote the development of EVCI [17].

In summary, related EVCI policy analysis have mostly focused on the relationship between EVCI and new energy vehicle adoption. In reality, the promotion and adoption of new energy vehicles seems to be a more important issue for governments. Many new energy vehicle policies have been launched to increase the uptake of new energy vehicles, including improvement of charging infrastructure and charging services, purchase subsidies, transportation privileges, tax incentives, and so on. In particular, research that focuses on policy instrumentation of Chinese EVCI policies has not been widely carried out. Thus, in this paper, we conduct policy content analysis to explain the characteristics of the policy instruments implemented by the Chinese government in promoting the development of the EVCI industry. This paper aims to enrich and expand the research field of policy content analysis, thus providing insight into China's EVCI policy concerns.

The remainder of this paper proceeds as follows. Section 3 outlines the analysis framework, Section 4 describes the analysis method and data, Section 5 discusses the numerical results, and Section 6 presents the conclusion and policy implications.

## 3. Theoretical Perspective and Analysis Framework

### 3.1. Theoretical Perspective

Public policy theory suggests that public policy is established by policy agents through an orderly choice of policy instruments that reflect their values [18]. The essence of a policy instrument is that it is an instrument adopted by the government to achieve a certain goal [19]. The government must choose its policy instruments (or combinations of them) scientifically and reasonably when formulating the policies, in order to effectively achieve its policy objectives [20]. From this perspective, policy instruments provide a key basis for policy research at the policy-maker level [21].

With the increasing complexity of public policy formulation and implementation in the practice and the continuous development of the policy discipline, the theory of policy instruments has gradually gained the attention of scholars. The related studies vary in their research perspectives and classification criteria of scholars. Rothwell and Zegveld

categorized policy instruments from supply, demand, and environmental perspectives. They argue that supply-side policy instruments are designed to directly drive industrial development, environmental policy instruments are designed to indirectly influence industry development, and demand-side policy instruments are designed to pull industry development [22]. This classification method focuses on the essential attributes of policy instruments and brings the scenarios of policy application into the research category. Therefore, in this paper, we borrow constructs from the instrumental typology of Rothwelland and Zegveld (supply-side policy instruments, environmental policy instruments, demand-side policy instruments), in order to develop a typology that differentiates instruments, depending on their impact on industry (see Table 1).

*3.2. Analysis Framework*

First, according to Porter's "Diamond model" theory, the government's public policies affect many aspects of industrial development [23]. The government can influence industry development through providing subsidies, financial support, education and training, and so on. They may also cultivate and guide consumer demand with government procurement and the formulation of relevant industrial standards, regulate the order of industry market with policies and regulations, and create a good development environment for industries. In fact, Porter's theory is in the same vein as Rothwell and Zegveld's idea. This paper takes policy instruments as one dimension of the EVCI policy analysis framework.

Secondly, the effects of policies are influenced by different stakeholders and, so, the policy maker must consider the operation and integration of all parts of industrial chain regarding the overall layout of policy instruments [24]. According to the different effects of policy instruments, they should be matched with industrial chain parts scientifically and reasonably for sustainable development of the industry. Therefore, as another analysis dimension, the EVCI industrial chain is included in the analysis framework in this paper. The EVCI industry chain can be divided into five parts, namely, manufacturing and maintenance, construction and operation, power supply, information platform service, and new energy vehicle market application. Manufacturing and maintenance includes the manufacturing and production of charging piles and other supporting facilities, as well as their maintenance after putting into operation. Construction and operation mainly includes the investment, construction, and operation of charging stations/piles, where the main body is the charging stations/piles construction operator. Power supply mainly includes the power and energy supply required for the construction and operation of charging stations/piles, and the main body is electric energy enterprises. Information platform service mainly includes building charging service platforms to provide integrated charging station/pile information service for new energy vehicle users, and its main body is the information platform service providers. New energy vehicle market application mainly includes the use of charging infrastructure in the new energy electric vehicle market, such as in private households, the unit and rental passenger car market, and the commercial vehicle market, as well as including public transportation, municipal sanitation, and logistics, in addition to electric vehicle time-sharing rentals.

The two-dimensional policy analysis framework of "policy instrument–industrial chain" constructed in this paper is summarized in Figure 1.

**Table 1.** Instrumental typology of EVCI policies.

| Instrumental Typology | Included Instruments | Definitions |
|---|---|---|
| Demand-side Instruments | Government Procurement | Centralized charging infrastructure and charging services purchasing with public funds provided by public institutions and organizations. |
| | Price Subsidy | Subsidies and tax incentives provided by government for EVCI industrial entities. |
| | Trade Control | Regulatory measures on foreign trade of charging infrastructure. |
| | Pilot Demonstration | The government develops pilot cities and demonstration projects, and improves the understanding of people. |
| Environmental Instruments | Administration Approval | The government simplifies approval procedures for charging infrastructure development. |
| | Organization Support | The government sets up special organizations and clarifies their responsibilities for charging infrastructure development. |
| | Intellectual Property Protection | The government clearly identifies and protects the rights of patent, trademark, product design for charging infrastructure development. |
| | Regulatory Control | The government issues a series of laws and regulations to restrict or maintain charging infrastructure market behavior and to create a favorable environment for people. |
| | Financial Support | The government expands investment and financing channels for charging infrastructure-related enterprises, and encourages innovative in business models. |
| | Goal Planning | An element of the charging infrastructure policies by setting a timetable and determining a plan to achieve the development goal. |
| Supply-side Instruments | Talent Training | Formulating long-term talent development measures for promoting charging infrastructure, which include increasing the number of practitioners and providing skills training. |
| | Public Service | Providing charging information connectivity mechanism, charging infrastructure information services. |
| | Fund | The government directly provides financial resources and special funds for charging infrastructure development. |
| | Technology Research and Development | Providing public scientific and technological support for charging infrastructure. |
| | Supporting Facilities | It mainly includes land and power grid and other facilities supporting charging infrastructure. |

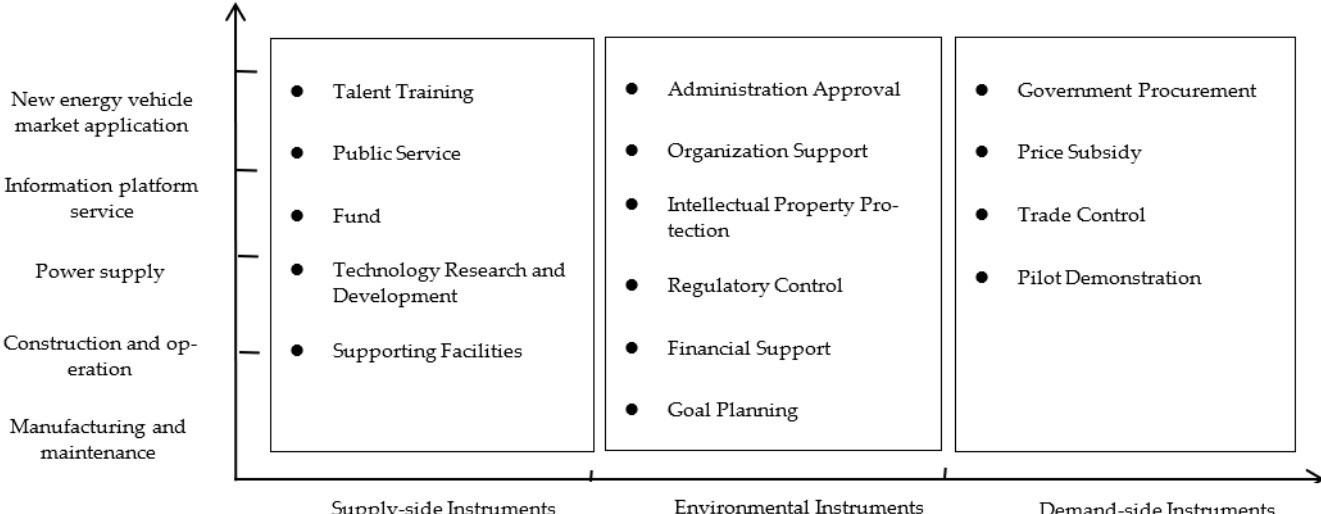

**Figure 1.** The two-dimensional policy analysis framework.

## 4. Research Methodology and Data

### 4.1. Quantitative Analysis of Policy Content

We used a quantitative content analysis approach to analyze the central policies directly related to electric vehicle charging infrastructure in China. Quantitative analysis of policy content transforms the unstructured content information contained in policy texts into measurable data by setting up categories and codes to analyze the key features of the policy [25]. It is essentially a semantic analysis method, effectively reducing the interference of subjectivity and allowing one to analyze policy literature in a systematic and standardized way [26]. The purpose of applying a quantitative approach to content analysis in this paper was to understand the text of the measures supporting the development of China's EVCI industry, as well as to clarify the policy instruments and policy concerns of the central government. The process of implementing the methodology was as follows: selecting the study sample, identifying the content analysis units, setting up the categories, coding, analyzing, and interpreting the result.

### 4.2. Data Collection

Our policy content analysis drew on data from current policies of the central government relevant to development of EVCI industry in China, covering the years ranging from 2014 to 2021. The data were mainly derived from the Peking University Law Database and government websites. EVCI policies were searched with "charging", "new energy vehicles", "electric vehicles", and "charging infrastructure" as subject terms, keywords, and titles, yielding a large number of policy texts. In order to make the research samples more typical and explanatory, the initial search samples were selected based on the following principles:

- Policies released during the period from 1 January 2014 to 31 May 2021;
- Policies formulated by the Party and the central government authorities, and their various functional departments;
- The content of policies must be highly relevant to the EVCI industry. A sample will be excluded if it is only briefly mentioned in other policies;
- The types of policy documents are mainly relevant laws, regulations, and ministerial decrees, excluding informal working documents, such as approvals and working letters.

A secondary search of government and ministerial websites was conducted, in order to ensure the completeness and validity of the policy samples. Then, we invited several experts in the field of EVCI to select the samples, and adjusted the selected result according to their opinions. Finally, 37 policy samples were selected for analysis.

### 4.3. Coding Scheme

The analysis unit in this paper was the 37 policy texts included in the analysis. The analysis categories were the policy instrument and the industrial chain dimensions of EVCI in the abovementioned two-dimensional policy analysis framework. By coding and categorizing the analysis units included in the policy text, in accordance with "Policy Number–Chapters–Specific Clauses", a policy text content analysis unit coding table was formed (see Table 2). Considering the article length, Table 2 only shows the content analysis units coding of the Number 1, 2, and 37 policy texts, while the content analysis units coding for the other policies are not displayed here. The coding of the No. 1 content analysis unit (i.e., "1-1-1") indicates that the first specific clause, namely, "In order to make the design of electric vehicle charging stations implement the relevant national guidelines and policies, unify technical requirements, achieve safety and reliability, advanced technology, economic and reasonable, this specification is formulated" is in the first chapter of the first policy document "Code for design of electric vehicle charging station." On one hand, in the policy instrument dimension, this analysis unit is coded as "Regulatory control," which belongs to the environmental instruments. On the other hand, in the industrial chain dimension, it is coded as "construction and operation of charging stations/piles."

It should be noted that, if more than one policy instrument is used in a unit, the count is repeated; if a policy article covers more than one industry chain part, the count is also repeated. Accordingly, if a unit does not specify that the measure is directed at a particular part of the industry chain, it is coded as "the whole chain."

**Table 2.** Content analysis units coding table of electric vehicle charging infrastructure policy text (partly).

| No. | Policy Text Name | Content Analysis Units | Coding | Policy Instrument | Industrial Chain Parts |
|---|---|---|---|---|---|
| 1 | Code for design of electric vehicle charging station | In order to make the design of electric vehicle charging stations implement the relevant national guidelines and policies, unify technical requirements, achieve safety and reliability, advanced technology, economic and reasonable, this specification is formulated. | 1-1-1 | Regulatory Control | Construction and operation of charging stations/piles |
| 2 | Implementation plan for the purchase of new energy vehicles by government agencies and public institutions | In accordance with the principle of "enterprise investment as the mainstay, the government is encouraging and guiding formation of joint efforts and active and steady promotion", we will fully mobilize the enthusiasm of all social parties, strengthen the construction of charging infrastructure for new energy automobile, guarantee the charging demand, and build a charging infrastructure and service system that meets the operational needs of new energy vehicles in line with the scale of use. | 2-2-2 | Goal Planning | Construction and operation of charging stations/piles |
| | | Local governments should incorporate new energy vehicle charging infrastructure into the overall planning of urban construction and development as urban public infrastructure in accordance with the principle of being moderately advanced and advancement and guaranteeing access to charging infrastructure. This includes introducing relevant policies to encourage full competition among qualified enterprises for the construction and operation and maintenance of new charging infrastructure. | 2-4-8 | Goal Planning | Construction and operation of charging stations/piles |
| | | For the new or renovated car parking lots of the government and public institutions, it should be taken into account the new energy vehicle equipment renewal plan with fully consideration of the charging needs, setting up special parking spaces for new energy vehicles and building charging piles and gradually increase the number of charging piles. | 2-4-9 | Government procurement | Manufacturing and maintenance of charging equipment; Construction and operation of charging stations/piles |
| | | To establish and standardize market access standards and encourage social capital to participate in new energy vehicle production and charging operation services. | 2-4-10 | Government procurement | Manufacturing and maintenance of charging equipment; Construction and operation of charging stations/piles |

**Table 2.** *Cont.*

| No. | Policy Text Name | Content Analysis Units | Coding | Policy Instrument | Industrial Chain Parts |
|---|---|---|---|---|---|
| 37 | Guidelines for Promoting Auto Consumption in the Commercial Sector and Some Local Experiences and Practices | All localities can give comprehensive incentives to consumers in the purchase of new energy vehicles, charging, traffic, parking, and other aspects. | 37-4-3 | Fund | New energy vehicle market application |
| | | Facilitate the charging (changing) of new energy vehicles, encourage places that are in a position to do so to introduce subsidies for the construction and operation of charging (changing) infrastructure, support the construction of charging (changing) infrastructure relying on gas stations, highway service areas, street lights, etc., and guide enterprises and institutions to build charging facilities at a rate of no less than 10% of the number of existing parking spaces. | 37-4-9 | Public Service Regulatory Control | Construction and operation of charging stations/piles |

## 5. Results and Discussion

### 5.1. Distribution Characteristics of EVCI Policy Texts

In terms of the text types, the 37 policies contained seven types, namely, guidance, implementation scheme, plans, program, guideline, notice, and standard (see Table 3). Policies including guidelines, implementation schemes, plans, program, and guidelines are mainly designated to guide the development of the EVCI industry. In general, these top design-level texts accounted for about 51% of the total, and generally include development goals, key tasks, organization support, and so on. Accordingly, the specific operational-level policy texts, namely, notices and standards, accounted for about 49% of the total. Therefore, the number of strategic policies and specific operational policies was relatively balanced. This result indicates that the Chinese government has not only issued many policies at the top design-level, but has also issued a series of policies at the implementation level to promote the development of the EVCI industry.

**Table 3.** Statistics of policy text types.

| Type | Guidance | Implementation Scheme | Plan | Program | Guideline | Notice | Standard | Total |
|---|---|---|---|---|---|---|---|---|
| Count | 4 | 4 | 6 | 4 | 1 | 13 | 5 | 37 |

### 5.2. Dimensional Analysis of EVCI Policy Instruments

Based on the coding results of policy texts, all content analysis units were further classified and statistically analyzed, and the content analysis results for the policy instruments and EVCI industrial chain were obtained (see Table 4).

The statistical results showed that the 37 EVCI policy texts used the policy instruments 415 times in total. Supply-side instruments were used 127 times, accounting for 30.6% of the total; environmental instruments were used 195 times, accounting for about 47.0%; and demand-side instruments were used 93 times, accounting for 22.4% (see Figure 2). These results reveal that the Chinese government has mainly used environmental instruments, followed by supply-side instruments, while demand-side instruments are rarely used. It is obvious that these instruments are used unevenly in the overall policy structure, and there is a large difference in proportion between them.

**Table 4.** Content analysis results of policy instruments and EVCI industrial chain.

| Instrument Type | Instrument Name | Manufacturing and Maintenance | Construction and Operation of Charging Stations/Piles | Power Supply | Information Platform Service | New Energy Vehicle Market Application | Total |
|---|---|---|---|---|---|---|---|
| Supply-side instruments | Talent training | 0 | 0 | 0 | 0 | 0 | 127 |
| | Public service | 6 | 15 | 7 | 12 | 13 | |
| | Fund | 3 | 15 | 6 | 3 | 7 | |
| | Technology research and development | 5 | 6 | 2 | 2 | 1 | |
| | Supporting facilities | 0 | 15 | 2 | 0 | 7 | |
| Environmental instruments | Administration approval | 0 | 5 | 0 | 0 | 0 | 195 |
| | Organization support | 3 | 16 | 3 | 0 | 2 | |
| | Intellectual property protection | 0 | 0 | 0 | 0 | 0 | |
| | Regulatory control | 19 | 54 | 15 | 10 | 19 | |
| | Financial support | 2 | 12 | 6 | 1 | 6 | |
| | Goal planning | 2 | 17 | 1 | 1 | 1 | |
| Demand-side instruments | Government procurement | 13 | 14 | 1 | 0 | 10 | 93 |
| | Price subsidy | 1 | 12 | 8 | 0 | 9 | |
| | Trade control | 0 | 0 | 0 | 0 | 0 | |
| | Pilot demonstration | 5 | 5 | 5 | 5 | 5 | |
| Total | | 59 | 186 | 56 | 34 | 80 | 415 |

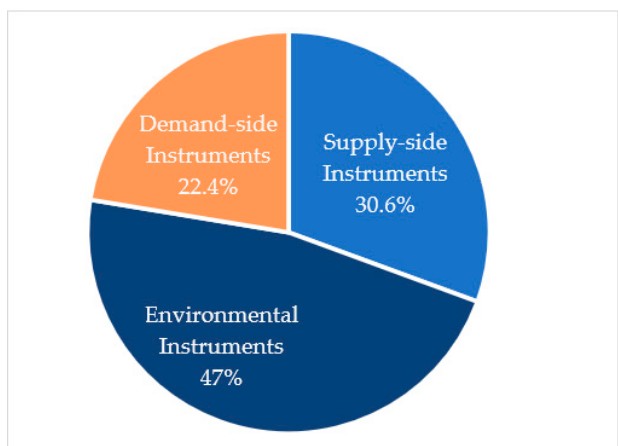

**Figure 2.** Frequency ratio of the three types of EVCI policy instruments.

Statistical results of specific policy instruments were obtained by combining the data in Table 4, as shown in Figure 3. In Figure 3, regulatory control was used 117 times: 53 times in public service, 38 times in government procurement, 30 times in price subsidy, 25 times in pilot demonstration, and 5 times in administration approval. Trade control, intellectual property protection, and talent training were all used 0 times, which means that

none of these three policy instruments appeared in any of the policy documents. However, the number of regulatory control is 117. This significant difference showed that policy instruments in EVCI policy documents were used unevenly. Meanwhile, it reflects that the Chinese government prefers to use coercive measures to manage the development of charging infrastructure, while neglecting trade control, intellectual property protection, and talent training. Therefore, it is obvious that the use of EVCI policy instruments has excessive, insufficient, or missing problems in different aspects and, so, the distribution is not balanced.

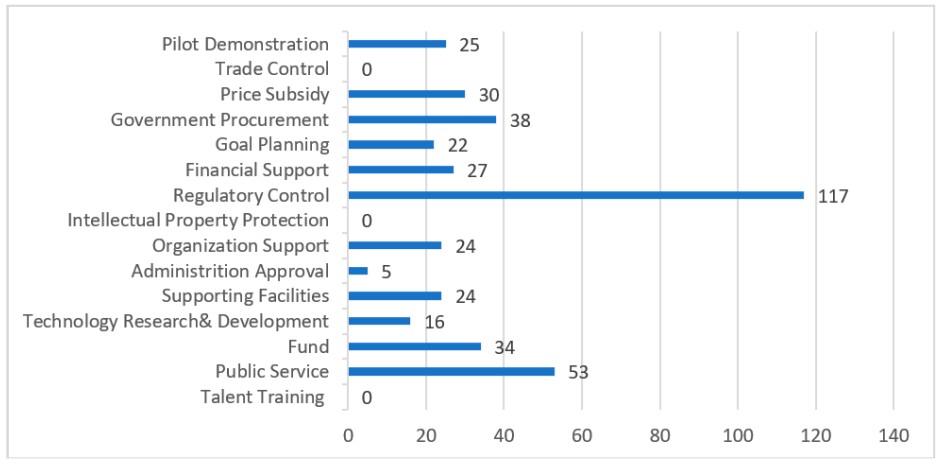

**Figure 3.** Frequency of EVCI policy instruments.

First, environmental instruments accounted for 47.0%, almost half of the total, indicating that the Chinese government has attached great importance to the construction and environment of the EVCI industry. Second, the internal structure of policy instruments is not balanced, and the frequency ratio of instruments is quite different. Among them, regulatory control was particularly prominent (117 times), accounting for 60.0% of the environmental instruments and for 28.2% of the total instruments. This indicates that the Chinese government is eager to build a good industrial environment by formulating industry standards, technical standards, and enterprise systems. In contrast, administrative approval was used only five times. This reveals that the procedure of EVCI construction and operation involves many departments and is not easy and convenient. In addition, intellectual property protection was not involved, which may limit the technological innovation in the EVCI industry in China.

Second, in terms of supply-side instruments, although the state has issued policies to support research and development into EVCI technologies, these are all vague guidelines, lacking specific initiatives for promotion.

Finally, demand-side instruments were used less than other policy instruments. Government procurement appeared most often, while the fact that trade control was not mentioned may indicate the difficulty in fully exploiting the pulling power of the EVCI industry.

### 5.3. Dimensional Analysis of EVCI Industry Chain

In the EVCI industrial chain, policy relating to charging stations/piles construction and operation occupied the main position, accounting for 44.8%. Market application of new energy vehicles, charging facility manufacturing and maintenance, power supply, and information platform service accounted for 19.3%, 14.2%, 13.5%, and 8.2%, respectively (see Figure 4). This demonstrates that, since 2014, the national policies of EVCI industry have mainly focused on the construction and operation of charging stations/piles and the market application of new energy vehicles, reflecting that China's EVCI industry is still in the initial development stage.

Through classified statistics of policy instruments, the distribution of three types of policy instruments corresponding to each part of EVCI industry chain could be obtained, as shown in Figure 5.

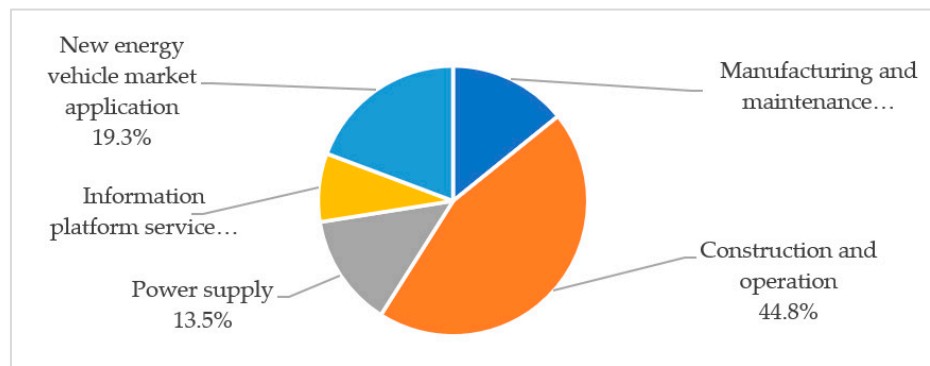

**Figure 4.** Frequency ratio of EVCI industrial chain.

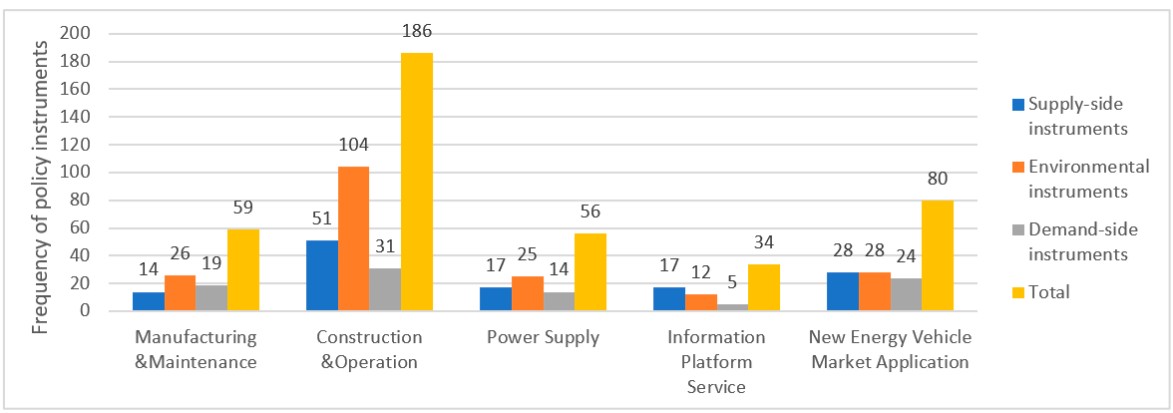

**Figure 5.** Distribution of policy instruments in EVCI industrial chain.

Combining the results shown in Table 4 and Figure 5, it is not difficult to see the following: (1) In terms of manufacturing and maintenance of EVCI, the environmental instruments are the most used, followed by demand-side and supply-side instruments. Specifically, representative instruments mainly include regulatory control and government procurement. (2) In terms of the construction and operation of charging stations/piles, environmental instruments are again the most used, followed by supply-side and, finally, demand-side instruments. Representative instruments mainly include regulatory control, goal planning, and organization support. (3) In terms of the power supply, environmental instruments are also the most used instrument, followed by supply-side and demand-side instruments. Representative instruments mainly include regulatory control, price subsidy, and public service. (4) In terms of information platform service, supply-side instruments are used the most, followed by environmental and demand-side instruments. Representative instruments mainly include public service and regulatory control. (5) In terms of the market application of new energy vehicles, the use ratio of the three types of instruments is relatively balanced. Representative instruments mainly include regulatory control, public service, and government procurement.

*5.4. Evolution Analysis of EVCI Policy*

5.4.1. Characteristics of EVCI Policy Evolution

Considering the lagged effect of policies, as well as the obvious differences in the number of policies issued and the use of instruments in each year, we only analyzed the proportion of policy instruments involved in the newly issued EVCI policies every year

and the proportion of the EVCI industry chain (see Figures 6 and 7). It should be noted that the data for 2021 were not included in the analysis, as the sample selection period used in this paper was up to 31 May, 2021, during which only one policy had been released by the government.

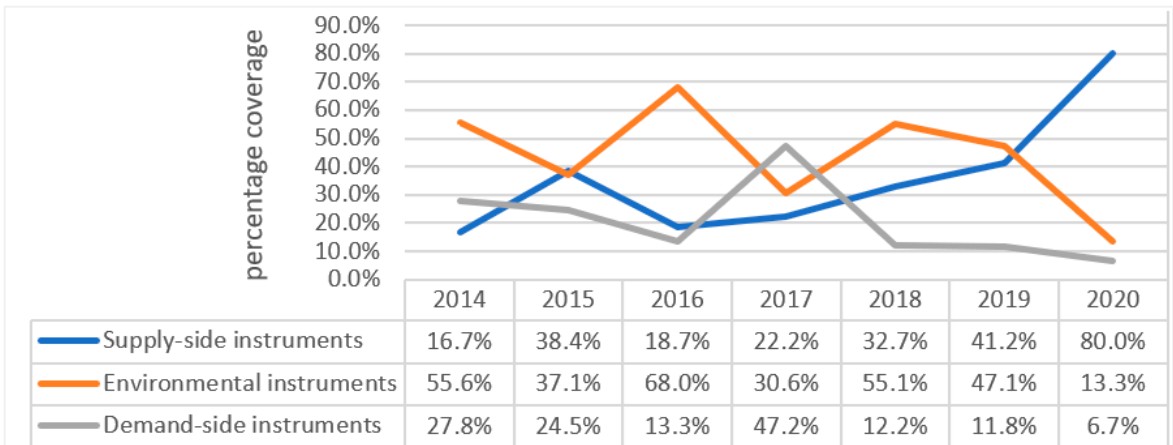

**Figure 6.** The use of policy instruments of newly issued EVCI policies over the years.

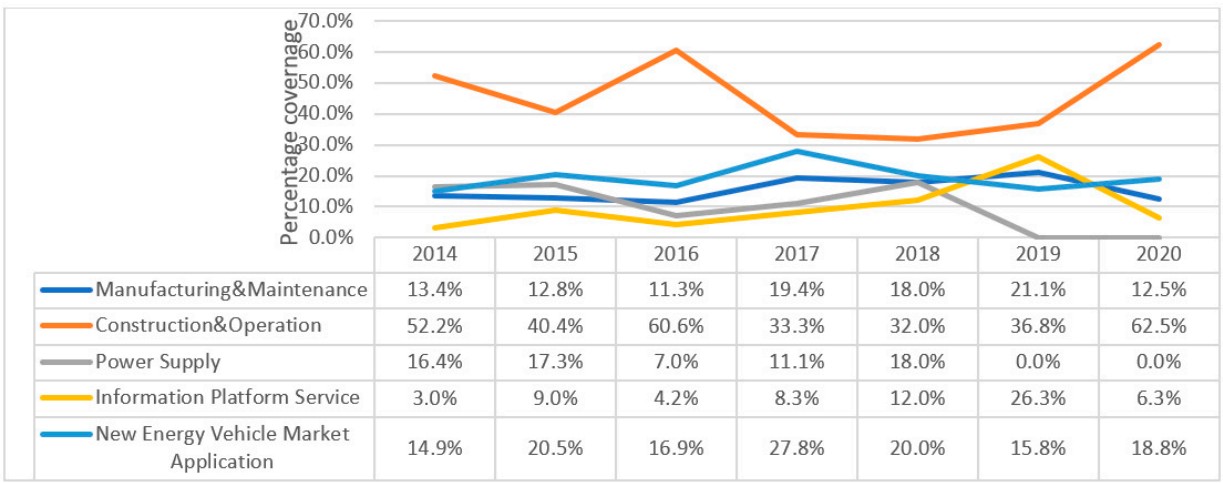

**Figure 7.** Distribution of EVCI industry chain supported by newly issued policies over years.

As can be seen from Figure 6, the supply-side instruments curve showed an upward trend in 2014–2015. This is related to a policy issued by China in 2014, "The Guidance on accelerating the promotion and application of new energy vehicles". In this policy, the Chinese government clearly proposed to increase financial support and improve public service by fostering a good market service and application environment. The continued rise in the curve after 2016 is due to the government's focus on the construction of EVCI and the supply of its public service in subsequent policies. For example, in 2016, "The notice on accelerating the construction of electric vehicle charging infrastructure in residential areas", which emphasizes on renovation of facilities in existing residential areas and construction of facilities in new residential areas, provided support for the site requirements of EVCI construction. In addition, "The implementation plan for the new national standard for electric vehicle charging infrastructure interface", which was also released in 2016, called for enhancing the publicity and training of the new national standard, as well as improving the product certification and access management system. This policy provided support for the development of EVCI, in terms of technology research and development.

Generally speaking, compared with the other two types of policy instruments, the proportion of environmental instruments was high. The environmental instruments curve



peaked in 2016, as almost all of the nine policies issued during this year referred to expanding investment and financing channels for EVCI companies, implementing charging price management policies, strengthening charging safety management, and clarifying the government departments and their responsibilities. This reflects the government's frequent use of environmental instruments, such as administrative approval, organization support, regulatory control, and financing support during this period. The downward trend after 2019 is due to the fact that the government had initially established a relatively comprehensive policy system by this time, and the policy focus is no longer needs to be on creating a friendly policy environment for the development of EVCI.

Although demand-side instruments were less frequently chosen than the other two types of policy instrument, the curve rose suddenly to 47% in 2017, even well ahead of the environmental instruments. This may be due to the fact that, in 2017, China released "The notice on accelerating the construction of electric vehicle charging infrastructure within government agencies", which set clear requirements for the proportion of parking lot charging infrastructure to be built within government agencies, and encouraged companies and government institutions by providing subsidies and a special state construction fund. In addition, the Chinese government has also initiated a nationwide demonstration project of EVCI built in public institutions and state-owned enterprises.

In summary, environmental and supply-side instruments are mainly used by the state in promoting the development of the EVCI industry. In contrast, demand-side policy instruments have been used less frequently and gradually declined.

### 5.4.2. Evolution Analysis of EVCI Industrial Chain Supported by Policy Instruments

In Figure 7, throughout the whole industrial chain, the support for the construction and operation of charging stations/piles was the greatest, which was always significantly higher than other parts. However, its fluctuation range was also large (the proportion decreased from 60.6% to 33.3% in 2016, and then increased significantly after a slow increase during 2017–2019). The change of support for the manufacturing and maintenance of charging facilities is consistent with the trend of the market application of new energy vehicles, which has been relatively stable. Support for power supply fluctuated within 20% during 2014–2018, and no longer appeared in 2019 and 2020. Support for information platform service was the lowest in 2014, but the overall trend was on the rise. It is obvious that the key point of national EVCI policies is mainly focused on the construction and operation of charging stations/piles. With the rapid growth of new energy vehicle sale, the quality requirements of users for charging services are increasing. Thus, the importance of information platform service was highlighted in all industrial chain parts. In recent years, the policy support of the state to the information platform service has gradually increased.

## 6. Conclusions and Policy Implications

### 6.1. Conclusions

In this paper, we adopted a policy content analysis approach to establish a policy instrument–industrial chain two-dimensional analysis framework, allowing us to conduct a systematic analysis of China's EVCI policies. Our research conclusions are as follows:

First, the current EVCI policy system in China has been fully established. To date, EVCI policies have consisted of strategic planning, regulations, industry standards, subsidies and incentives, and so on. Among them, top-level design policies accounted for 51%, and specific implementation policies accounted for 49%. Thus, the allocation of the policies is balanced and the whole system tends to be perfect.

Second, China's EVCI policy instruments were used unevenly, mainly reflected in the following aspects: first, demand-side instruments are in seriously short usage, supply-side instruments are also significantly under exploited, and environmental instruments are overused; second, there is a clear preference for the use of coercive instruments, such as regulatory control. This indicates that the state aims to regulate the EVCI industry market by forcing policy executors and policy targets to obey government requirements, while

lacking attention to intellectual property protection; meanwhile, public service, financial support, and supporting facilities are frequently picked up. The specific measures for R&D are insufficient at present. As for demand-side instruments, although the use of other policy instruments except trade control is relatively balanced, their overall use is relatively insufficient.

Third, the layout of policy instruments in the EVCI industrial chain needs to be improved. Chinese national policies have focused on the construction and operation of EVCI, and the state has mentioned building charging intelligent service platforms, establishing information interconnection mechanism, and standardizing and unifying industry standards in many policies; however, support for information platform service still accounts for the lowest percentage in the entire industrial chain.

Finally, the evolution of China's EVCI policies can be characterized by a decline in fluctuations in environmental instruments, an increase in fluctuations in supply-side instruments, and fluctuations in the use of a lower range of demand-side instruments. The construction and operation of EVCI obviously is the core object, which has always received more attention than other parts, while power supply and information platform service both require further attention and support.

*6.2. Policy Implications*

The way forward to strengthening the management and policy instrumentation of the central Chinese government to support the realization of good EVCI governance and new energy vehicle development can be considered as follows:

In terms of policy instrumentation, the use of demand-side instruments needs to be increased; for example, by formulating more specific laws and regulations for technologies and patents of EVCI and new energy vehicle development, strengthening the protection of intellectual property rights, and supplementing measures for the lack of trade control. The government also needs to further enhance the use of supply-side instruments by issuing more detailed support measures for the related research and development of technologies, increasing the number of people working in relevant fields, and providing skill training for enterprises.

In terms of the EVCI industrial chain, the government should pay more attention to power supply and information platform service in future developments. With regard to electricity supply, on one hand, the government should focus on supporting the research and application of new technologies, such as vehicle–grid interactions, and support power grid enterprises to create innovation platforms for the integration of new energy vehicles and smart energy in conjunction with vehicle enterprises. In addition, it also should carry out cross-industry joint innovation and R&D, encouraging the promotion of intelligent and orderly charging. On the other hand, the government should explore the implementation path for new energy vehicles to participate in the electricity spot market, as well as studying and improving the trading and scheduling mechanisms for new energy vehicle consumption and the storage of green electricity. As for information platform service, they should be integrated with big data, Internet of Vehicles, Internet of Things, smart grid, cloud computing, and other advanced technologies, in order to build an intelligent charging service network of "new energy vehicles–charging infrastructure–energy network–intelligent information service platform". Therefore, more scientific and reasonable policies need to be formulated and implemented. The government should also accelerate the establishment of connectivity mechanisms for charging operating company platforms to achieve information sharing, as well as multi–channel payment and settlement. These initiatives can greatly improve charging convenience and user experience. Meanwhile, innovation in business models, such as the integration of parking and charging, in order to achieve the interconnection of parking and charging data information, also need to be encouraged, as well as implementing measures to benefit the public, such as parking discounts for charging vehicles.

Although we used policy content analysis in this paper to reduce the shortcomings of qualitative research, which is susceptible to subjective influences, the actual effects of policy implementation still need to be further investigated through field research and specific cases. In addition, after the national policies on charging infrastructure were released, local governments responded and formulated a series of EVCI policies successively. Actually, these are the main force promoting the development of China's charging infrastructure industry. In this context, what are the characteristics and differences of local government policies, in terms of policy tool selection, institutional setting, and network relations? What is the difference between their behavioral logic and policy practice? What is the diffusion effect of policy among different levels of government? These questions are worth researching in depth in future studies.

**Author Contributions:** Conceptualization, X.W.; methodology, X.W.; software, X.W.; validation, X.W.; formal analysis, X.W.; investigation, X.W.; resources, X.W.; data curation, G.L.; writing—original draft preparation, X.W.; writing—review and editing, J.W. and K.Z.; visualization, X.W.; supervision, C.X.; project administration, J.W. and C.X.; funding acquisition, J.W. and C.X. All authors have read and agreed to the published version of the manuscript.

**Funding:** This research was funded by Innovation Method Fund of China (grant number 2018IM020300; 2019IM020200), Joint Funds of the National Natural Science Foundation of China (grant number U1904210-4), National Natural Science Foundation of China (grant number 71702172), Ministry of Education, Humanities and Social Science Projects (grant number 17YJC630183).

**Institutional Review Board Statement:** Not applicable.

**Informed Consent Statement:** Not applicable.

**Data Availability Statement:** Not applicable.

**Conflicts of Interest:** The authors declare no conflict of interest.

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
