# Peer review of "Electric Vehicle Charging Infrastructure Policy Analysis in China: A Framework of Policy Instrumentation and Industrial Chain"

_sustainability, doi:10.3390/su15032663_

Round 1
Reviewer 1 Report
The comments are given in the attachments.

Reviewer 2 Report
1. In page number 9 Table -2 place check the policy number. The policy number suddenly moving to 37 from 2. Please assign the correct numbering.
2. The effectiveness of various policy instruments can be projected according to various seasons and the power supply production basis considering both conventional and non conventional energy sources.
3. The policy instruments can be further classified based on urban and rural areas power consumption and also based on geographical advantages.
Reviewer 3 Report
The document should be shortened to around 10 pages. In fact, the instruments and their interpretation should be presented in a more concise way avoiding repetitions and redundancies.
Round 2
Reviewer 3 Report
The authors have improved their publication